# The Occurrence of *Apiognomonia hystrix* and Its Pathogenicity towards *Acer pseudoplatanus* and *Fraxinus excelsior* under Field Conditions

Tadeusz Kowalski, Piotr Bilański *[ID] and Bartłomiej Grad

Department of Forest Ecosystems Protection, University of Agriculture in Krakow, Al. 29 Listopada 46, 31-425 Krakow, Poland; rltkowal@cyf-kr.edu.pl (T.K.); bartlomiej.grad@urk.edu.pl (B.G.)
* Correspondence: piotr.bilanski@urk.edu.pl; Tel.: +48-1266-253-69

**Abstract:** *Apiognomonia hystrix* is an ascomycetous fungus within Diaporthales that is found on maples and to a lesser extent on other hardwood trees in Europe, Northern America and Asia. To date, varying opinions on the species' status as a cause of plant diseases have been expressed. In this study, we present the results of analyses conducted from 2012–2017 at forest sites in Poland on the occurrence of *A. hystrix* on *Acer pseudoplatanus* and *Fraxinus excelsior* and the pathogenicity of this fungus towards both tree species. For the sycamore leaves, *A. hystrix* conidiomata were detected in connection with 19.2% of galls caused by *Dasineura vitrina*, 20.4% of galls caused by *Drisina glutinosa* and 67.9% of extensive vein-associated necroses. The *A. hystrix* colonization of galls caused by both midge species resulted in statistically significantly larger necroses. On European ash leaves, conidiomata of *A. hystrix* occurred in connection with 0.8% of *Dasineura fraxinea* galls. Perithecia of *A. hystrix* were detected on overwintered leaf petioles in 8.1% of *A. pseudoplatanus* and 1.2% of *F. excelsior* samples. Twelve representative cultures were characterized molecularly by barcoding three marker genes (ITS, ACT, CAL). Results of phylogenetic analyses indicate that *A. hystrix* isolates are genetically variable, and three lineages are distinguishable. Eight isolates, including four originating from sycamore and four from European ash, were used to determine *A. hystrix* pathogenicity. Among the 48 *A. pseudoplatanus* petioles inoculated with *A. hystrix*, 41 developed necrotic lesions after 8 weeks, with the average necrosis length caused by particular isolates ranging from 14.5 to 67.2 mm. None of the 48 inoculated *F. excelsior* petioles developed necrotic lesions. Finally, selected aspects of *A. hystrix* morphology on natural substrates and in vitro are discussed in this paper, as well as the species' potential to cause disease symptoms.

**Keywords:** Gnomoniaceae; disease symptoms; anthracnose; ash dieback; gall midges





## 1. Introduction

*Apiognomonia hystrix* (Tode) Sogonov is an ascomycetous fungus within the Diaporthales order that occurs in Europe, Northern America and Asia [1–9]. A long-standing taxonomic classification of the species called for its division into two species depending on the substrate: occurrences on woody substrates were described as *Cryptodiaporthe hystrix* [10–13], while occurrences on leaves were described as *Gnomonia cerastis* [2,5,13–16]. This was justified by some morphological differences in the structure of the ascomata that developed depending on the substrate, i.e., the occurrence of a rudimentary stroma and the formation of grouped perithecia that appear to be more common in individuals growing on woody substrates [8]. However, this variation is well within the limits of the re-evaluated concept of the genus *Apiognomonia*, and *A. hystrix* was determined to be congeneric with *Apiognomonia*-type species, i.e., *A. veneta* in multigene phylogenetic analysis [8,17]. Nevertheless, both species show considerable variation in their anamorphic stages. The *A. veneta* anamorph, previously described as *Discula nervisequa*, produces unicellular elliptical conidia [17], while conidia of *Diplodina acerina*, the anamorph of

*A. hystrix*, are hyaline and one-septated [18]. Teleomorphs of both species may be found on overwintered leaves, petioles and dead tissue on twigs, while their respective anamorphic stages also occur within necroses on living leaves [5,17]. Both species produce abundant conidiomata and conidia in cultures in vitro [8,17]. Unlike *A. hystrix*, *A. veneta* produces an additional step within its developmental cycle, i.e., the spermatial stage described as *Gloeosporidina platani* [19].

Species of *Apiognomonia* exhibit a range of host specificity tendencies. Some species, such as *A. errabunda*, occur on a variety of plant families, while others, such as *A. acerina* or *A. erythrostroma*, are highly specialized to one genus of host plant, *Acer* and *Prunus*, respectively [8,13,20,21].

*Apiognomonia hystrix* occurs predominantly on *Acer* spp. infecting leaves, twigs and buds [1,6,13,14,16,18,22,23]. Locally, it was also detected in the trunks of sycamore (*Acer pseudoplatanus*) trees with declining symptoms [24–26]. The species is known to form a variety of trophic interactions with host plants. *Apiognomonia hystrix* is a very common endophyte in living symptomless leaves of *A. pseudoplatanus* [6,22,23,27], but also occurs in necrotic areas of *A. pseudoplatanus* leaves, surrounding midge galls caused by *Dasineura vitrina* [6,22] and *Drisina glutinosa* [27]. In fact, *A. hystrix* is an important factor contributing to the high mortality of larvae inhabiting these galls [5,6,22]. It has been demonstrated that *A. hystrix* is also a frequent endophyte in living branches of sycamore trees [28]. Similar results were obtained in an endophyte survey of large leaf maples (*A. macrophyllum*) in Canada, where only infrequent occurrence of *A. hystrix* was detected in leaves, but the species was a predominant endophyte in twigs [4,29].

Analyses of *A. hystrix* pathogenicity performed to date have produced ambiguous results. According to Wulf [22], necroses caused by *A. hystrix* surrounding *Da. vitrina* galls are limited in size and do not expand with time, suggesting that the fungus lacks pathogenic potential. Kowalski [27] describes not only necroses surrounding *Dr. glutinosa* galls but also long vein-associated necroses of *A. hystrix* with abundant conidiomata. Sieber et al. [4], on the other hand, reports no disease symptoms on twigs of *A. macrophyllum* despite frequent occurrence of the fungus. Numerous plant pathological textbooks (e.g., [30–32]) do not mention *A. hystrix* as a causal agent of any disease, while others (e.g., [33]) do indicate its potential to cause diseases. According to Monod [13], *A. hystrix* occasionally causes round necroses, with a diameter of 1 to 2 cm, on leaves of *Acer* spp. Butin [34] recognizes the species as a causal agent of round or irregularly shaped leaf spots, while according to Brandenburger [35], the spots caused by *A. hystrix* are irregularly shaped and start to develop from the tip or edges of the leaves. The pathological significance of *A. hystrix* towards the leaves of sycamore trees has not been investigated to date, and the only analysis of *A. hystrix* pathogenicity involving this host was on its trunks [25].

In addition to maples, *A. hystrix* occurs on various other hardwoods [8,13], but high occurrence is only observed in some of these species. An example of such a case is the dominant position of *A. hystrix* among endophytes of *Quercus gambelii* [15]. In leaves of *Q. grisea* and *Q. gambelii* hybrids, the frequency of *A. hystrix* was lowest on hosts resembling *Q. grisea* and increased towards those resembling *Q. gambelii* [36]. For the leaves of these oaks, a positive correlation between *A. hystrix* frequency and larval mortality of the leaf-mining moth *Phyllonorycter* sp. was observed [15,36]. Monod [13] reported that *A. hystrix* (as *G. cerastis*) has been detected on species from nine genera of plants, although for many of these species, observations were infrequent. However, *A. hystrix* has recently been repeatedly reported as an endophyte of *Fraxinus excelsior* (European ash) leaves and twigs [37–39].

For over two decades, ash dieback, an emerging fungal disease caused by the ascomycete *Hymenoscyphus fraxineus,* has been present in Poland [40–43]. During our studies on *F. excelsior* endophytes and mycobiota associated with ash leaf petioles in litter, *A. hystrix* was detected for the first time in Poland in this particular host species [44,45]. Following this detection, additional observations were performed regarding the prevalence of

*A. hystrix*, in connection with galls *Da. vitrina* and *Dr. glutinosa* on leaves of *A. pseudoplatanus*, and *Drisina fraxinea* on leaves of *F. excelsior*.

In all three cases, galls are produced by larvae. *Dasineura vitrina* larvae produce small pustule galls, 4–7 mm in diameter mainly on nerves of the leaves. As an effect, numerous holes are produced in the leaf blade. *Drisina glutinosa* larvae reside in a droplet of liquid and produce small depressions on the lower leaf surface, 1 mm across and deep, surrounded by a pale zone 5–7 mm in diameter. The upper surface develops an indistinct discoloured bulge. Larvae of both species develop in galls on leaves for 2–5 weeks, and then they drop to the soil where they hibernate. *Dasineura fraxinea* causes an inconspicuous circular blister, 6–8 mm across, slightly raised on the upper surface, with irregular rounded openings on the underside, which are green or yellowish at first, then brown after the larva has left [46,47].

Thus, using the collection of *A. hystrix* isolates acquired in other experiments, we designed the described study, with the primary aim to determine the pathogenic potential of *A. hystrix* towards *A. pseudoplatanus* and *F. excelsior* by artificial inoculations of leaf petioles on living trees in the field. Additional aims of the study were as follows: (i) description of disease symptoms associated with *A. hystrix* infection, (ii) morphological characterization of fruiting bodies in vivo and in agar-grown cultures, and (iii) detailed molecular and phylogenetic characterization of isolates originating from *F. excelsior* and *A. pseudoplatanus* coupled with providing high-quality reference data to the NCBI GenBank database.

## 2. Materials and Methods

### 2.1. Occurrence of A. hystrix Conidiomata on Leaves with Galls and on Vein-Associated Necroses

In this experiment, we analysed 400 *F. excelsior* leaflets with necroses caused by *Dasineura fraxinea*, 100 *A. pseudoplatanus* leaves with galls caused by *Dasineura vitrina* and 100 *A. pseudoplatanus* leaves with galls caused by *Drisina glutinosa* (Table 1) were used. Five to ten leaves per site were collected in young forest stands located mostly in southern Poland (Supplementary Material Figure S1) from July to September (sporadically up to mid-October) in the 2012–2017 period. All of the leaves were alive during collection but showed characteristic damage caused by the respective gall midge species. The leaves collected at each site were packed into sterilized paper envelopes (B4). Identification of galls was performed by Prof. M. Skrzypczyńska, mostly as a part of a wider monograph on galls and the midges that cause them [48]. A total of 50 *A. pseudoplatanus* leaves and 10 leaflets of *F. excelsior* with extensive vein-associated necroses (henceforward referred to as anthracnose) were collected at the same locations (Table 1).

The first step of laboratory analyses involved separately counting all instances of damage associated with the gall causing midges for each of the analysed insect species. Then, the size of necroses was estimated for each leaf (measurements with a ruler under a stereomicroscope, accuracy 1 mm). In case of irregularly shaped necroses surrounding midge galls, it was estimated as an average of length and maximum width. Necrotic lesions with length ≥20 mm was treated as an anthracnose (Table 1). Conidiomata of *A. hystrix* occurring within necroses were identified morphologically using the literature sources [5,13,18,34]. The conidiomata-grown conidia were collected and used to initiate representative cultures for downstream analyses.

**Table 1.** Occurrence of *Apiognomonia hystrix* conidiomata in relation to gall midge species and within necrotic areas of *Acer pseudoplatanus* and *Fraxinus excelsior* leaves, analysed from 2012–2017.

| Tree Species [1] | Gall Midge Species | Damage | Analysed Leaves/Leaflets (N) | Analysed Necrotic Lesion (N) | Number (%) of Necrotic Lesion Carrying *A. hystrix* Conidiomata |
|---|---|---|---|---|---|
| *Acer pseudoplatanus* | *Dasineura vitrina* | galls | 100 | 704 | 135 (19.2) |
| | *Drisina glutinosa* | galls | 100 | 851 | 174 (20.4) |
| | | anthracnose [2] | 50 | 56 | 38 (67.9) |
| *Fraxinus excelsior* | *Dasineura fraxinea* | galls | 400 | 494 | 4 (0.8) |
| | | anthracnose | 10 | 10 | 0 (0.0) |

[1] Sampling sites are shown in Supplementary Material Figure S1. [2] anthracnose—vein-associated necroses.

### 2.2. Occurrence of A. hystrix Ascomata on Leaf Petioles from the Previous Year

To determine the occurrence of *A. hystrix* on leaf petioles, we used the data previously collected during our extensive mycological investigations of fungal species related to ash dieback as well as diseases of other species of trees. The data encompass the results of the mycological analyses of 2700 previous year leaf petioles of *F. excelsior* (Table 2) that were conducted in the period of 2012–2019. The term 'petiole', in relation to ash leaves, is used here to indicate the entire main axis of an ash leaf, including the distal rachis. The petioles originated from 23 forest stands, approximately 30 to 120 years old (Supplementary Material Figure S2), in which 20 to 60 randomly selected petioles (2–6 petioles from 10 locations within a stand), were sampled. Most often, the stands were sampled only once, but multiple sampling per stand (3 to 8 times) was performed in six stands. For 10 stands, an additional 860 *A. pseudoplatanus* petioles were collected from the leaf litter (Supplementary Material Figure S2 and Table 2). All petioles were examined for the presence of fungi whose species were determined in the laboratory based on the morphology of fruiting bodies. *Apiognomonia hystrix* colonization was recognized based on the characteristic traits of its ascomata [5,8,12,13]. This analysis enabled us to determine the occurrence of *A. hystrix* ascomata on leaf petioles of *F. excelsior* and *A. pseudoplatanus*. Representative petioles colonized by *A. hystrix* were used to start cultures for downstream analyses.

**Table 2.** Occurrence of *Apiognomonia hystrix* ascomata on overwintered leaf petioles of *Fraxinus excelsior* and *Acer pseudoplatanus*.

| Tree Species | Analysed Season [1] | Sampling Year | Number of Sampling Sites (Sites with Confirmed Occurrence of *A. hystrix*) [2] | Number of Analyses Petioles | Number (%) of Petioles with Ascomata of *A. hystrix* |
|---|---|---|---|---|---|
| *Fraxinus excelsior* | a | 2013–2017 | 9 (4) | 820 | 10 (1.2) |
| | b | 2013–2019 | 23 (0) | 1880 | 0 (0.0) |
| | total | | 23 (4) | 2700 | 10 (0.4) |
| *Acer pseudoplatanus* | a | 2013–2019 | 9 (5) | 270 | 22 (8.1) |
| | b | 2013–2017 | 10 (1) | 590 | 1 (0.2) |
| | total | | 10 (5) | 860 | 23 (2.7) |

[1] Petioles analysed in April–August (a), September–December (b). [2] Sampling sites are shown in Supplementary Material Figure S2.

### 2.3. Fungal Isolations and Morphological Observations

*Apiognomonia hystrix* cultures that were necessary for further experiments were acquired by direct isolation from conidia, produced in conidiomata, occurring within necroses or by isolation from petiole sections. All isolations and subsequent in vitro culturing steps were conducted using 2% malt extract agar medium (MEA: 20 g L$^{-1}$ malt extract, Difco, 15 g L$^{-1}$ agar; Difco, Sparks, MD, USA, supplemented with 200 mg L$^{-1}$ tetracycline; Tetracyclinum TZF Polfa, Poland) in Petri dishes, at 20 °C in the dark. Droplets of conidial mass erupting from conidiomata were collected with a sterile scalpel from necrotic fragments of leaf blades that had been incubated in moist conditions (15–18 h, 5 °C) and the droplets were spread over the surface of MEA medium. When germination started, six to ten small pieces of MEA with germinating conidiospores were cut out and transferred onto fresh Petri dishes with MEA medium. Isolations of pure cultures from petioles colonized by *A. hystrix* in the previous year involved surface disinfection of plant material using the ethanol/sodium hypochlorite method following the protocol described by Kowalski et al. [49]. The same procedure was used to acquire a few *A. hystrix* cultures from living asymptomatic petioles and shoots and from dead current year petioles of *F. excelsior* (Table 3). Twelve randomly selected isolates of *A. hystrix*, acquired from various substrates, were DNA sequenced using several barcode fragments and were subsequently used in phylogenetic analysis (Table 3). Representative fungal strains are currently stored on MEA slants at 5 °C at the culture collection of the Department of Forest Ecosystems Protection, University of Agriculture in Kraków (Table 3), whereas all specimens carrying *A. hystrix* conidiomata and ascomata are stored dried at the department's herbarium.

Fungal structures were observed in distilled water on microscope slides. Microscopic features were measured using a Zeiss V12 Discovery stereomicroscope and a Zeiss Axiophot

light microscope using differential interference contrast illumination. Photomicrographs were taken with AxioCam MRc5 and HR3 digital cameras (Zeiss, Göttingen, Germany).

For most of the structures (conidiomata, perithecia, asci, ascospores, microconidia), the dimensions were estimated based on at least 30 measurements, with minimum and maximum values reported. For galls and macroconidia, either in situ or in vitro, arithmetic means and standard deviations were reported with numbers of measurements given in parentheses. Thus, measurements are provided as min-max × min-max (mean ± S.D. × mean ± S.D. μm, *n* = x). Images were processed with GIMP 2.8.4 [50].

**Table 3.** Fungal isolates of *Apiognomonia hystrix* obtained in the present study.

| Isolate [1] | Substrate | Sampling Location | Collection Data | ITS | CAL | ACT |
|---|---|---|---|---|---|---|
| Fe852F [2] | living asymptomatic shoot | Miechów | 8 October 2016 | MW822184 | MW822196 | MW822208 |
| Ap954F [2] | living leaf with anthracnose | Stary Sącz | 31 August 2013 | MW822185 | MW822197 | MW822209 |
| Fe955F [2] | overwintered petiole in the litter | Kowary | 22 April 2013 | MW822186 | MW822198 | MW822210 |
| Fe167E | overwintered petiole in the litter | Domiarki | 3 May 2014 | MW822187 | MW822199 | MW822211 |
| Fe168E [2] | leaf with gall of *Da. fraxinea* | Ojców | 12 August 2014 | MW822188 | MW822200 | MW822212 |
| Ap176E [2] | overwintered petiole in the litter | Kowary | 24 June 2014 | MW822189 | MW822201 | MW822213 |
| Ap229E [2] | living leaf with gall of *Dr. glutinosa* | Nowy Targ | 23 July 2014 | MW822190 | MW822202 | MW822214 |
| Fe853F | dead petiole | Miechów | 18 October 2016 | MW822191 | MW822203 | MW822215 |
| Fe876F [2] | living asymptomatic petiole | Miechów | 11 August 2016 | MW822192 | MW822204 | MW822216 |
| Ap877F [2] | living leaf with gall of *Da. vitrina* | Ojców | 1 July 2014 | MW822193 | MW822205 | MW822217 |
| Ap878F | living leaf with gall of *Da. vitrina* | Ojców | 1 July 2014 | MW822194 | MW822206 | MW822218 |
| Ap879F | living leaf with gall of *Dr. glutinosa* | Ojców | 1 July 2014 | MW822195 | MW822207 | MW822219 |

[1] The first two letters indicate the host species: Fe—*Fraxinus excelsior*, Ap—*Acer pseudoplatanus*. [2] Isolates used in temperature assay and in pathogenicity test.

### 2.4. Cardinal Growth Temperatures

The temperature assay was performed using eight cultures, including four from each host tree (Table 3). Mycelial plugs, with a diameter of 8 mm from the edge of actively growing 21-day-old colonies of *A. hystrix*, were transferred onto 2% MEA in Petri dishes and incubated at 5, 10, 15, 20, 25, 30 and 35 °C in the dark. Two replicate plates were prepared for each fungal strain and temperature combination. After 10 days, the diameters of the resulting colonies were measured with a millimetre ruler, and two replicates were used to calculate the average colony diameter for each strain and temperature combination.

### 2.5. DNA Extraction, PCR and Sequencing

Genomic DNA was extracted from 3-week-old MEA-grown cultures using the Genomic Mini AX Plant Kit (A&A Biotechnology, Gdynia, Poland), according to the manufacturer's protocol. Three loci, 18S-ITS1-5.8S-ITS2-28S (ITS rDNA), actin (ACT) and calmodulin (CAL), were amplified for sequencing and phylogenetic analyses using the following primers: ITS1-F [51] and ITS4 [52] or ITS5 and ITS4 [52] for ITS, ACT-512F and degenerate ACT-783R for ACT, CAL-228F and CAL-737R for CAL [53]. All gene fragments were amplified in a 25-μL reaction mixture containing 0.25 μL of Phusion Green High-Fidelity DNA polymerase (Thermo Fisher Scientific), 5 μL of Phusion Green HF buffer (5x), 0.5 μL of dNTPs (10 mM each), 0.75 μL of DMSO (100%), 0.5 μL of each primer (25 μM), 1 μL of DNA extract (20–100 ng/μL) and 16.5 μL of sterile deionized water. The cycling profiles included a denaturation step at 98 °C for 30 s followed by 35 cycles of 5 s at 98 °C, 10 s at 52–64 °C (depending on the optimal Tm of the primers) and 30 s at 72 °C, and a final chain elongation step at 72 °C for 8 min. All reactions were run using a LabCycler thermocycler (SensoQuest Biomedical Electronics GmbH, Göttingen, Germany). The PCR products were visualized under UV light in a 2% agarose gel stained with Midori Green DNA Stain (Nippon Genetic Europe, Düren, Germany).

Amplified products were sequenced bi-directionally using a BigDye® Terminator v 3.1 Cycle Sequencing Kit (Applied Biosystems, Foster City, CA, USA) and an ABI PRISM 3100 Genetic Analyser (Applied Biosystems, Foster City, CA, USA) at the DNA Research Centre (Poznań, Poland) with the use of PCR primers. All newly obtained sequences were deposited in GenBank (http://www.ncbi.nlm.nih.gov, accessed on 26 March 2021) with the accession numbers presented in Table 3.

The obtained sequences were used as queries in searches using the BLASTn [54] algorithm to retrieve similar sequences from GenBank for phylogenetic analysis (http://www.ncbi.nlm.nih.gov, accessed on 26 March 2021). For species and genus level, the accepted similarity sequences threshold was 99–100% and 91–98%, respectively. Accession numbers of the sequences retrieved are provided in Table 3.

Concatenated ITS-ACT-CAL sequences of *Apiognomonia* spp. with selected species of Gnomoniaceae were used in the phylogenetic analyses with *Cryphonectria parasitica* included as an outgroup. The dataset was compiled and edited in BioEdit v.2.7.5 [55] and aligned using the online version of MAFFT ver. 7 [56] with the following settings: the E-INS-i strategy with a 200 PAM/κ = 2 scoring matrix, a gap opening penalty of 1.53 and an offset value of 0.00. The alignment was checked manually with BioEdit v.2.7.5 [55] and compared with gene maps [57] to ensure that introns and exons were aligned properly.

Three different phylogenetic analyses were performed using maximum likelihood (ML), maximum parsimony (MP) and Bayesian inference (BI). The best-fitted substitution model was determined for ML and BI using the corrected Akaike Information Criterion (AICc) in jModelTest 2.1.10 [58,59], resulting in the GTR+G models being selected for the combined ITS-ACT-CAL dataset.

ML analysis was conducted with PhyML 3.0 [60] via the Montpelier online server (http://www.atgc-montpellier.fr/phyml/, accessed on 26 March 2021) using 1000 bootstrap pseudoreplicates to calculate node support values. MP analysis was conducted with PAUP* 4.0b10 [61]. Gaps were treated as fifth state characters. One thousand bootstrap pseudoreplicates were generated and analysed to determine the levels of confidence for the nodes within the inferred tree topologies. Tree bisection and reconnection (TBR) was selected as the branch swapping option. Tree length (TL), consistency index (CI), retention index (RI), homoplasy index (HI) and rescaled consistency index (RC) were recorded after the trees were generated. BI analysis based on a Markov chain Monte Carlo (MCMC) method was conducted with MrBayes v3.1.2 [62]. The chains were run for 10 million generations using the best-fitted model. Trees were sampled every 100 generations, resulting in 100,000 trees from both parallel runs. The default burnin, first 25% of samples, was used. The remaining trees were used to generate a majority rule consensus tree and to determine the posterior probability node support values. The results of the phylogenetic analyses were combined and visualized using TreeGraph 2.10.1-641 beta [63] and FigTree v1.4.0 [64]. Both the alignment and the trees generated in this study were deposited in the TreeBASE phylogenetic data repository (http://purl.org/phylo/treebase/phylows/study/TB2:S28003, accessed on 26 March 2021).

### 2.6. Pathogenicity Test

Pathogenicity tests were conducted at the experimental plot located in the Stary Sącz Forest District, southern Poland (49°33′83″ N, 20°39′35″ E), on eight- to ten-year-old *F. excelsior* and *A. pseudoplatanus* saplings. Inoculum production, inoculation and re-isolation were performed according to the methods described by Kowalski et al. [65]. The inoculum was inserted into superficial tissue incisions (7 mm long, prepared with a sterile scalpel) and covered with Parafilm. For inoculum production, colonies of *A. hystrix* were grown for ten days in darkness on MEA at 20 °C. Subsequently, small sterile sycamore petiole sticks (approximately 5 mm × 2 mm × 2 mm) were placed on the colonies and incubated for two additional weeks. Eight isolates, four originating from ash and four from maple, were used in inoculations (Table 3). Each of the isolates, regardless of their origin, was used to inoculate 6 petioles of *A. pseudoplatanus* and 6 petioles of *F. excelsior* saplings. In total, 48 petioles of each host tree were inoculated using 8 fungal isolates. Mock inoculations were performed for 6 petioles of each tree species. All inoculations were performed in mid-July 2017. The lengths of superficially visible necrotic lesions were measured, and the occurrence of fungal fructification was recorded after 8 weeks. Re-isolations were attempted from all inoculated and control petioles within 24 h of harvesting. Three to six tissue pieces were taken from either the inoculation wound, the advanced necrosis or the

lesion edge. Re-isolations were considered positive if the tested fungus grew from at least one tissue sample. In total, 360 samples from inoculated petioles and 36 samples from control plants were evaluated.

### 2.7. Statistical Analysis

The effects of temperature on the in vitro growth of *A. hystrix* isolates originating from *F. excelsior* and *A. pseudoplatanus* were analysed with the Kruskal–Wallis test, followed by nonparametric multiple comparison of mean ranks. The statistical significance of differences between the two groups within the analysed variables were determined using the Mann–Whitney U test. All statistical calculations were performed using STATISTICA software, version 12 [66].

## 3. Results

### 3.1. Occurrence of Galls and Apiognomonia hystrix on Leaves and Their Associated Symptoms

The number of *Da. vitrina* or *Dr. glutinosa* galls on the *A. pseudoplatanus* leaves analysed ranged from 1 to 94 per single leaf (Figure 1a,b). The vast majority of galls caused by these two species of midges were round (Figure 1a–d). Less frequently (2.6%), necrotic areas surrounding the galls were irregular in shape, reaching sizes of 9–18 × 6–12 mm (Figure 1e). The size of gall associated necroses depended on the occurrence of *A. hystrix*. The diameter of necrotic lesions associated with *Da. Vitrina* galls ranged from 4.0 to 15.0 (7.8 ± 1.8 mm, $n = 135$) or from 4.0 to 12.0 mm (6.5 ± 1.2 mm, $n = 569$), with and without co-occurring *A. hystrix*, respectively. Analogically, the diameter of necroses on *A. pseudoplatanus* leaves surrounding *Dr. glutinosa* galls ranged from 4.0 to 15.0 mm (7.8 ± 2.0 mm, $n = 174$) or from 4.0 to 10.0 mm (5.8 ± 1.1 mm, $n = 677$). Thus, the *A. hystrix* colonization of necroses caused by both species of gall midges resulted in them being statistically significantly larger (Figure 2). The higher numbers of galls occurring on single leaves resulted in the frequent merger of their associated lesions. For some of the leaves, the damaged sections of the leaf blade were more extended as a result of rapid development of necroses along the veins (Figure 1f,g). This type of necrosis reached a size of 20–53 × 6–12 mm. Among them, 46.4% did not occur in connection with galls (Figure 1i,j and Table 1). One of the secondary symptoms observed on the leaves with galls involved dieback of entire sectors of the leaf blade (Figure 1h). Conidiomata of *A. hystrix* developed in connection with 19.2% of galls due to *Da. vitrina*, 20.4% of galls due to *Dr. glutinosa* and 67.9% of vein-associated necroses (Table 1). Conidiomata occurred either on the leaf blade and/or on veins (Figure 1k,l), mostly on the lower side of the leaves but sporadically (<3%) on their upper side (Figure 1c). Initially, necrotic tissues were rusty-brown but turned light beige to light grey with time and as the conidiomata developed (Figure 1a–l). Occasionally, the affected part of the leaf blade did not noticeably turn brown, resulting in blurred edges of the necrosis. Such necroses could be distinguished from the healthy leaf blades by the presence of *A. hystrix* conidiomata (Figure 1m). A variety of fungal species other than *A. hystrix* have been found to occur in connection with galls on *A. pseudoplatanus* leaves comprising species of the genera *Alternaria*, *Cladosporium*, *Colletotrichum*, *Cristulariella*, *Discula* and *Phomopsis*.

*Fraxinus excelsior* necroses surrounding *Da. fraxinea* galls were analysed on 400 leaflets (Figure 1n and Table 1). Mostly, the necroses were round, with a diameter of 5–11 mm (mean 7.0 ± 1.3, $n = 149$), whereas 6.1% of the lesions had a less regular shape and size of 8–14 × 6–9 mm. This observation was mostly due to the limiting effect of leaf veins on necrosis growth. Only 0.8% of necroses surrounding *Da. fraxinea* galls carried *A. hystrix* conidiomata (Table 1), of which all but one occurred on the lower side of the leaflets. Initially, the damaged tissue sections were intensely brown (Figure 1m), while in late July and August, they turned light beige (Figure 1o). No *A. hystrix* was detected within large vein-associated necroses on *F. excelsior* leaves (Table 1). Similar to *A. hystrix* on *A. pseudoplatanus* leaves, a number of other fungi have been found to occur in connection with *Da. fraxinea* galls, including species of the genera *Alternaria*, *Cladosporium*, *Colletotrichum*, *Phoma*, *Phomopsis* and *Venturia*.

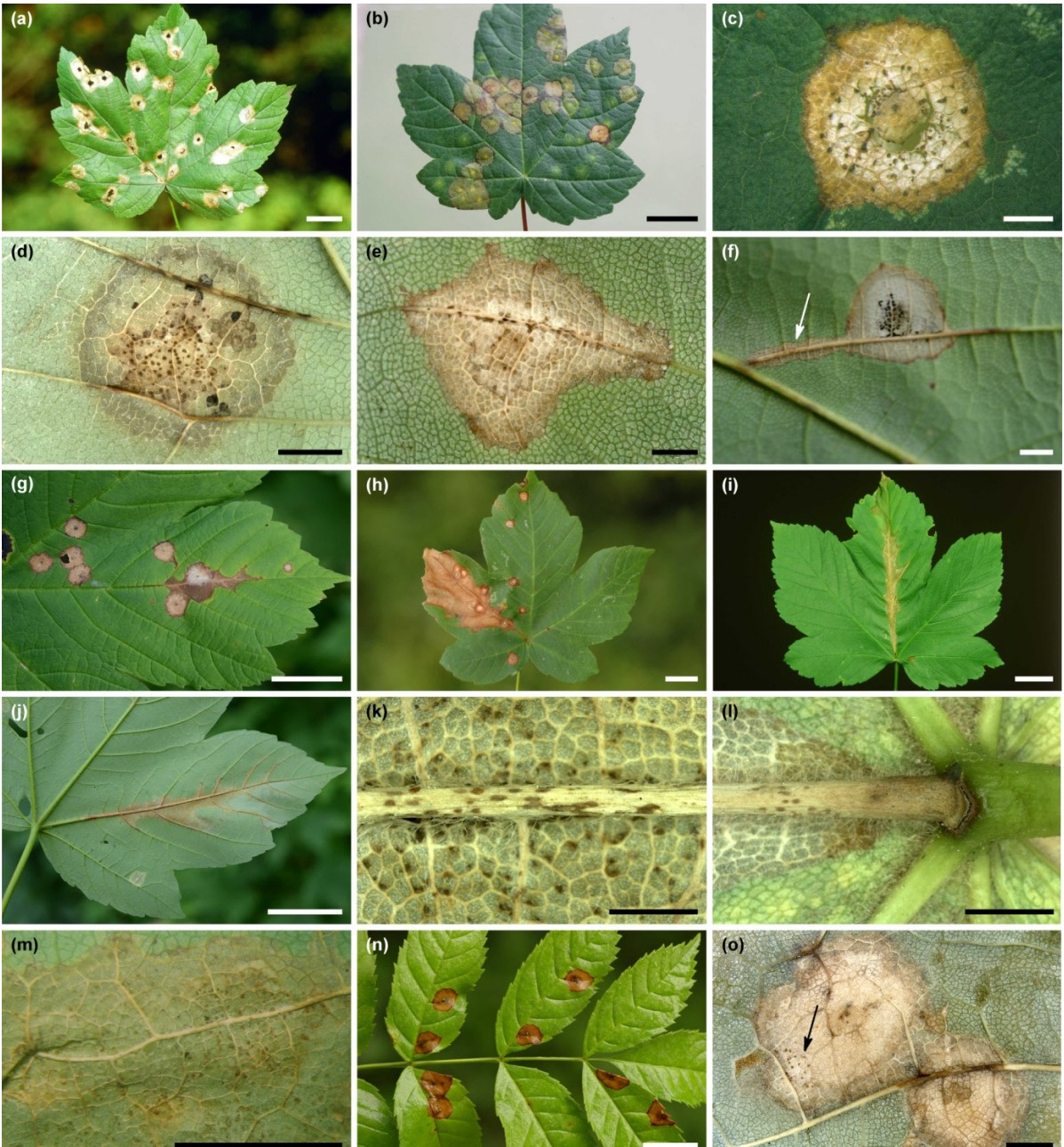

**Figure 1.** Galls, natural necrotic lesions and conidiomata of *Apiognomonia hystrix* on living leaves of *Acer pseudoplatanus* and *Fraxinus excelsior*. *Acer pseudoplatanus* (**a–m**): galls and necrotic lesions associated with *Dasineura vitrina* (**a**) and *Drisina glutinosa* (**b**); conidiomata of *A. hystrix* associated with *Da. vitrina*, upper side (**d**) and *Dr. glutinosa*, lower side of the leaf (**d,e**); anthracnose associated with *Da. vitrina* (**f**) and *Dr. glutinosa* (**g**); galls of *Dr. glutinosa* surrounded by necrotic leaf section (**h**); anthracnose on leaves associated with *A. hystrix*, no insect-related damage (**i,j**); conidiomata of *A. hystrix* on necrotic part of the leaf (**k–m**); *Fraxinus excelsior* (**n,o**): necrotic lesions associated with *Dasineura fraxinea* (**n**); conidiomata of *A. hystrix* on lower side of the leaf associated with galls of *Da. fraxinea* (**o**); scale bars: (**a,b**) = 20 mm, (**c–f**) = 2 mm, (**g–j**) = 20 mm, (**k,l**) = 2 mm, (**m,n**) = 20 mm, (**o**) = 2 mm.

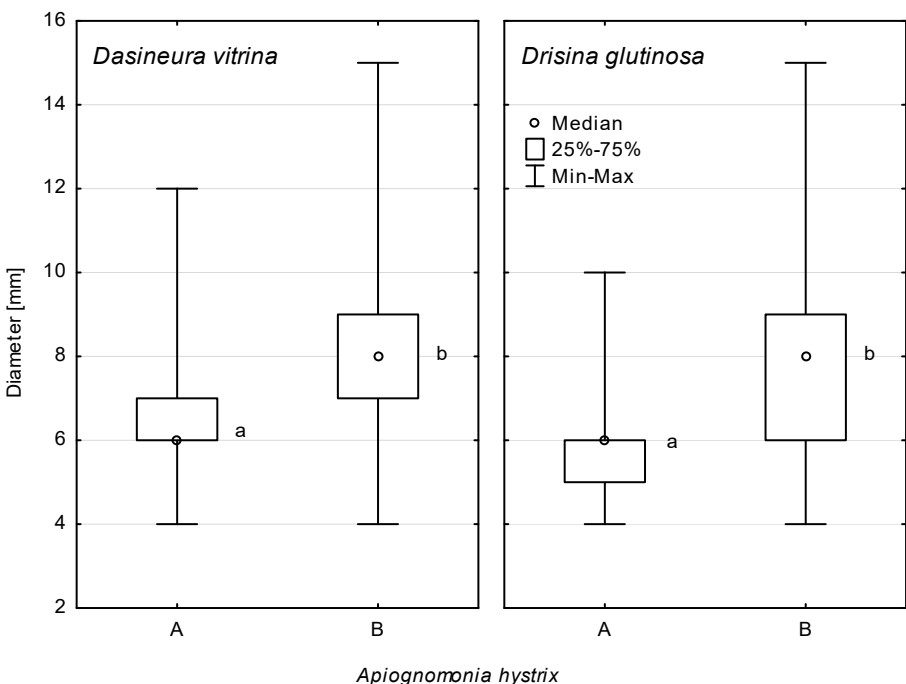

**Figure 2.** Diameters of necrotic lesions on leaves of *Acer pseudoplatanus* inhabited by *Dasineura vitrina* and *Drisina glutinosa* with (B) or without (A) co-occurrence of *Apiognomonia hystrix* infection. Values indicated with the same letters have been determined significantly not different at α = 0.05 in Mann–Whitney U test.

*Apiognomonia hystrix* perithecia were detected on 1.2% of overwintered leaf petioles of *F. excelsior* sampled in the April–August season (Figure 3a and Table 2), occurring in four out of 23 research plots sampled (Supplementary Material Figure S2 and Table 2). The perithecia were more frequently found on *A. pseudoplatanus*, occurring on 8.1% of the analysed petioles (Figure 3b,c and Table 2) and in half of the analysed research plots (Supplementary Material Figure S2 and Table 2). An example of a heavily colonized *A. pseudoplatanus* petiole harbouring over 80 perithecia is shown in Figure 3b. Ascomata of *A. hystrix* on overwintered petioles collected between September and December were detected only occasionally (Table 2).

Notably, along with abundant conidiomata, *A. hystrix* perithecia were also observed to develop in high numbers on several *A. pseudoplatanus* leaves collected in early October. The perithecia occurred both on the leaf blades and on the veins (Figure 3d).

### 3.2. Morphological and Phylogenetic Aspects

Perithecia developed on leaf petioles are globose or subglobose, blackish brown, immersed in dead tissue and are solitary to gregarious (Figure 3a–c). After disintegration and exfoliation of the peripheral layers of petioles, the perithecia may seem as if they were developed on the petiole surface (Figure 3c). Perithecia measured 200–500 × 200–460 μm in size. Perithecial necks, located centrally, are usually slightly curved, occasionally laterally flattened, 320–570 μm long, 40–60 μm wide at the base and 25–50 μm wide at the tip. The apical section of the neck, 90–130 μm long, is almost hyaline (Figure 3c). Asci are clavate, with a distinct apical structure, tapering below into a short stipe, 36–52 × 7.5–11.2 μm, 8-spored (Figure 3e). Ascospores, cylindrical-fusiform, two-celled, 15–18 × 2.2–3.2 μm, hyaline, slightly constricted at the single median septum, contain several small oil droplets in each cell. In some ascospores, appendages can be observed (Figure 3f).

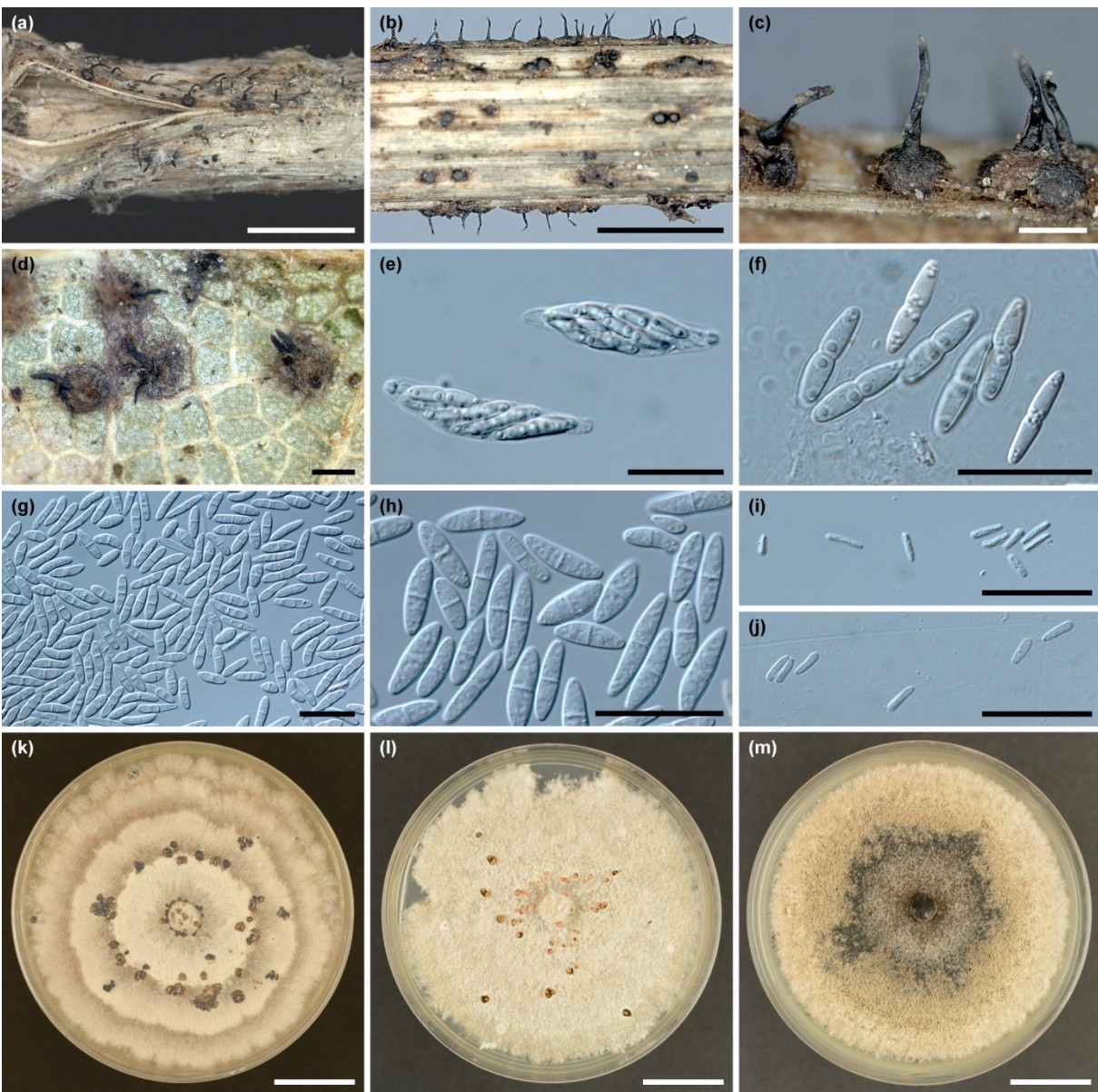

**Figure 3.** Morphology of *Apiognomonia hystrix* on natural substrates and in cultures. Perithecia on overwintered petioles of *F. excelsior* (**a**) and *A. pseudoplatanus* (**b**,**c**), perithecia on *A. pseudoplatanus* leaf, before falling (**d**), asci (**e**) and ascospores (**f**) from perithecium on petiole of *A. pseudoplatanus* in vivo, conidia developed in conidiomata on *A. pseudoplatanus* leaf in vivo (**g**), conidia from conidiomata in vitro (**h**), microconidia developed on *A. pseudoplatanus* leaf in vivo (**i**) and in conidiomata in vitro (**j**), colonies on MEA with varying structure and pigmentation, visible conidial matrix (20 °C, 3 weeks) (**k**–**m**); scale bars: (**a**,**b**) = 2 mm, (**c**,**d**) = 200 µm, (**e**–**j**) = 20 µm, (**k**–**m**) = 20 mm.

Conidiomata produced on necrotic areas of leaf blade were sub-peridermal, separate, visible as brown nodules, with a diameter of 0.15–0.6 mm (Figure 1c,d), while conidiomata on leaf veins were often more elongated, with a diameter of 0.2–0.8 × 0.2–0.6 mm (Figure 1k,l). Conidiophores were branched at the base, cylindrical, narrowing to the tip, 12–23 × 2.7–3.5 µm. Conidia were hyaline, fusiform, 1-septate, straight or slightly curved 10.0–19.0 × 2.5–4.5 µm (mean = 13.27 ± 2.28 × 3.11 ± 0.38, *n* = 90) (Figure 3g). Microconidia were single-celled, hyaline, club-shaped, 4–7 × 1.2–1.5 µm (Figure 3i). Microconidia were observed at the end of the growing season together with macroconidia or in separate conidiomata. For comparison, both types of in vitro-grown conidia are presented in Figure 3h,j.

Colonies of *A. hystrix* on MEA grew relatively quickly at 20 °C, reaching 5.8 to 7.3 cm after 10 days. They showed some level of morphological variation, depending on the isolate and culturing temperature. Most often, they were whitish with light-olive-grey tinges, with prominent concentric zones formed by rings of more abundant woolly mycelium (Figure 3k). Margin clear, even to lobate, reverse slightly darker compared to the colony surface. Hyphae hyaline to light olive, with abundant granulation, densely septated, 3 to 8 μm wide, with swellings up to 12 μm. A number of colonies showed a more uniform structure with flat, loose to shortly woolly aerial mycelium (Figure 3l). In some cultures, abundant light-brown to black patches, merging with each other, developed from the centre of the colony (Figure 3m). The abovementioned features were not stable within the isolates. Particular isolates inoculated into new dishes from culture-grown conidia or from colony fragments could sometimes develop colonies of a different type than the original isolate.

Abundant nodular conidiomata developed on the colony surface, which, at maturity, produced creamy to salmon slimy conidial drops (Figure 3k–m). In vitro grown conidia produced two-celled, fusiform, straight or curved, 8–23 × 2.2–4.5 μm (mean = 14.63 ± 3.61 × 3.24 ± 0.55 μm, *n* = 90) (Figure 3h).

Culture-grown macroconidia differed in size from those obtained directly from necroses on leaves of both host species. They were statistically significantly longer than macroconidia produced on necroses (Figure 4a). Likewise, the width of culture-grown macroconidia was greater than that of macroconidia in vivo, but the difference was not statistically significant (Figure 4b). In cultures older than 8 weeks, abundant 1-celled, club-shaped microconidia of 4–7 × 1.3–1.5 (1.8) μm were produced (Figure 3j).

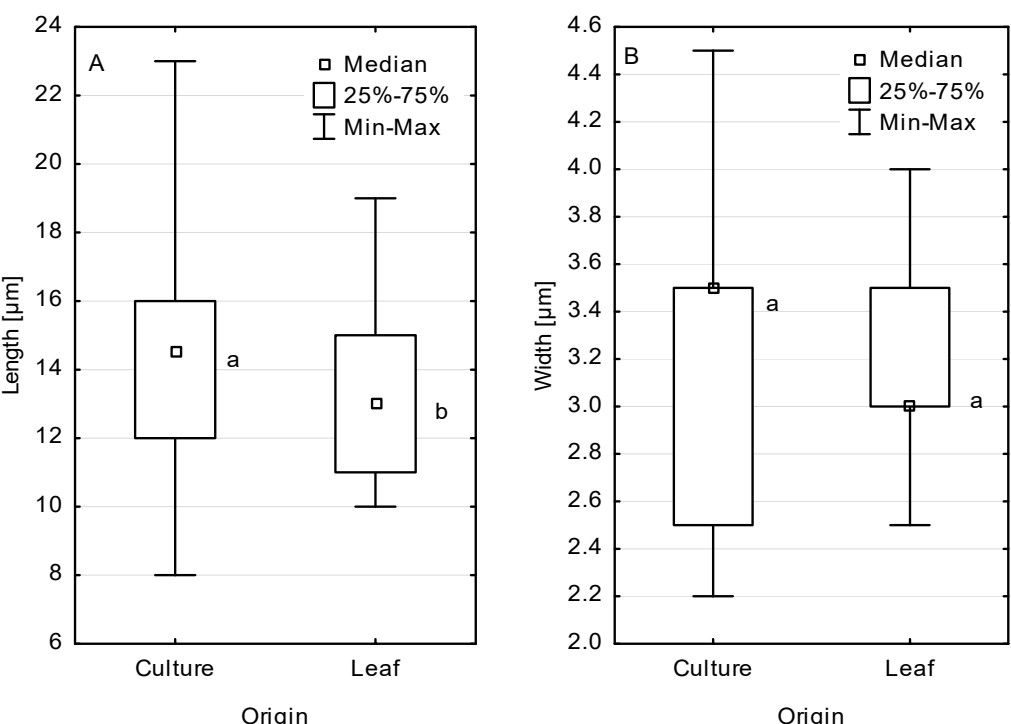

**Figure 4.** Length (**A**) and width (**B**) of *Apiognomonia hystrix* macroconidia produced in cultures and in necroses on leaves. Groups indicated with different letters in Mann–Whitney U test are significantly different at α = 0.05.

The analysed ITS sequences of *A. hystrix* isolates were grouped in two clades (Figure 5). However, all the isolates generated in this study belonged to a single clade, together with a number of reference isolates such as BPI 877696, BPI 877698 and CBS 100566. The other clade, sister to the previous one, comprised three isolates: CBS 910.79, CBS 911.79 and CBS 109759 (Figure 5), and was supported in two out of three (ML and MP) phylogenetic methods. Variability of ITS sequences was insufficient to differentiate *A. errabunda* and

*A. veneta* isolates that formed a single strongly supported clade, easily distinguished from *A. hystrix*.

The analysis of concatenated ITS-ACT-CAL sequences resulted in a more detailed tree (Figure 6). Here the analysed *A. hystrix* isolates formed three clades, designated lineages I, II and III. All the isolates acquired in this study, regardless of their host origin, grouped with the isolate Ap_KA_N02_III_PA_F02 in the lineage I. This was the only strongly supported lineage within the *A. hystrix* group. Lineage II comprises sequences of isolates similar to *A. hystrix* Ap_CZ_N01_PA_A03, obtained from *A. pseudoplatanus* by other authors. However, linage III comprises a single isolate CBS 911.79 (Figure 6), which in the ITS analysis was joined by two additional isolates. Both these isolates originated from *A. pseudoplatanus* and could not be included in the three-gene analysis due to the lack of their ACT and CAL sequences in the GenBank.

In the three-gene phylogeny, the most prominent clade within the Apiognomonia group contains two subclades, separating *A. errabunda* and *A. veneta* isolates. Both the *A. errabunda*/*A. veneta* clade and the *A. errabunda* and *A. veneta* subclades received very high node support values in all three phylogenetic analyses.

### 3.3. Apiognomonia hystrix Growth in Relation to Temperature

Both *F. excelsior*- and *A. pseudoplatanus*-originating isolates of *A. hystrix* showed similar patterns of growth reactions at various temperatures (Figure 7). In the temperature range of 5–20 °C, the colony diameter increased with temperature, but the differences between colony diameters at 10, 15 and 20 °C were not statistically significant. The optimal growth rate of *A. hystrix* colonies was recorded at 20 °C, as the colony diameters at higher temperatures rapidly decreased. Only one isolate (Ap176E) was able to grow at 30 and 35 °C. The average diameter of colonies growing at 20 °C was statistically higher than that of colonies growing at 5 °C and at temperatures from 25 to 35 °C (Figure 7).

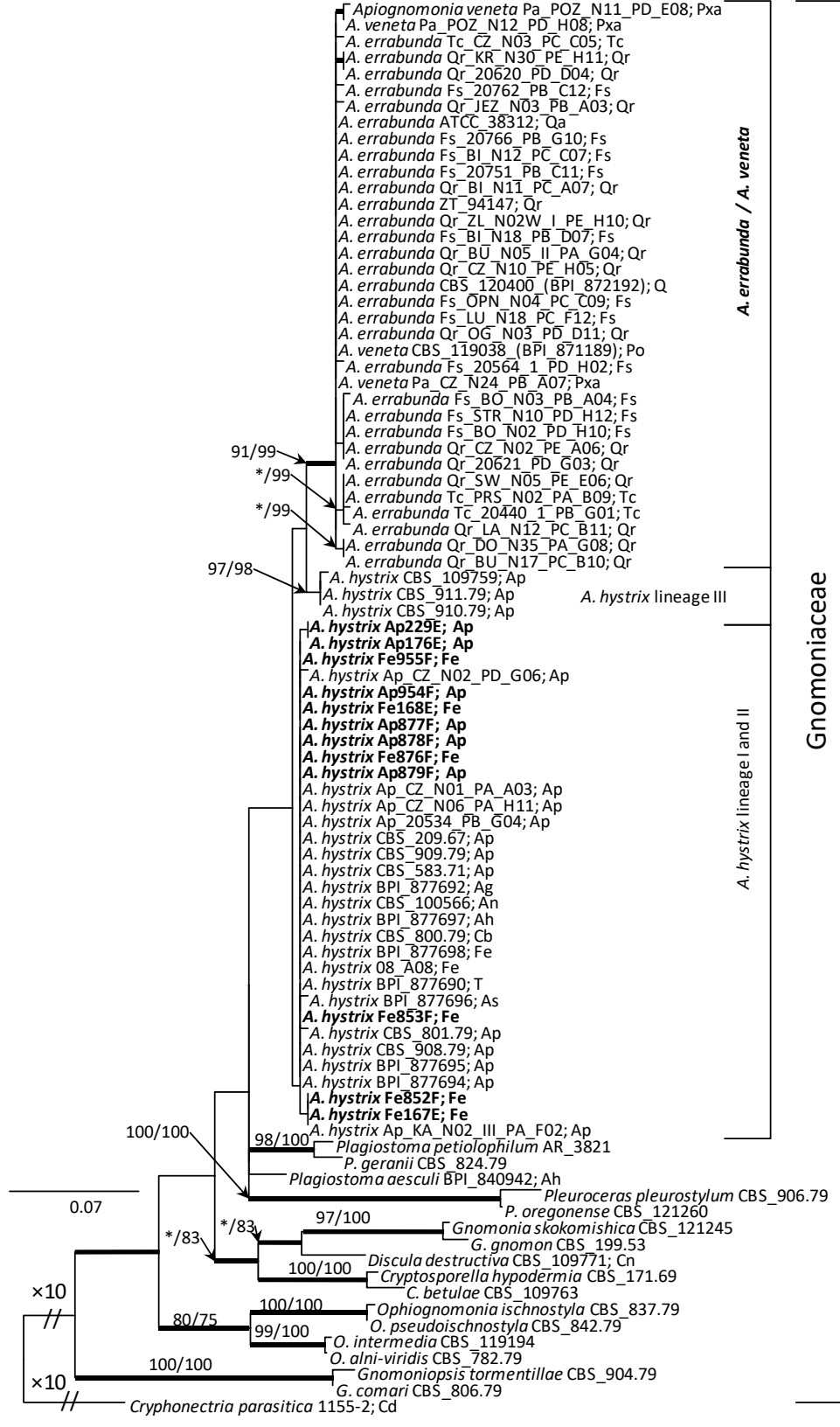

**Figure 5.** Phylogram obtained in maximum likelihood (ML) analysis of the ITS dataset of the Apiognomonia spp. and selected species from Gnomoniaceae. Sequences obtained in this study are marked in bold type. The bootstrap values ≥ 75% for ML and maximum parsimony (MP) analyses are presented at the nodes using ML/MP pattern. Bold branches indicate posterior probability values ≥ 0.95 obtained in Bayesian Inference (BI) analysis. * Bootstrap values < 75%. The tree is

drawn to scale (see bar) with branch length measured in the number of substitutions per site. *Cryphonectria parasitica* 1155-2 represents the outgroup. The abbreviations used after the semicolon indicate the host plants for isolates: Ah—*Aesculus hippocastanum*, Ap—*Acer pseudoplatanus*, Cd—*Castanea dentata*, Cn—*Cornus nuttallii*, Fe—*Fraxinus excelsior*, Fs—*Fagus sylvatica*, Po—*Platanus occidentalis*, Pxa—*Platanus × acerifolia*, Q—*Quercus* sp., Qa—*Quercus alba*, Qr—*Quercus robur*, Tc—*Tilia cordata*.

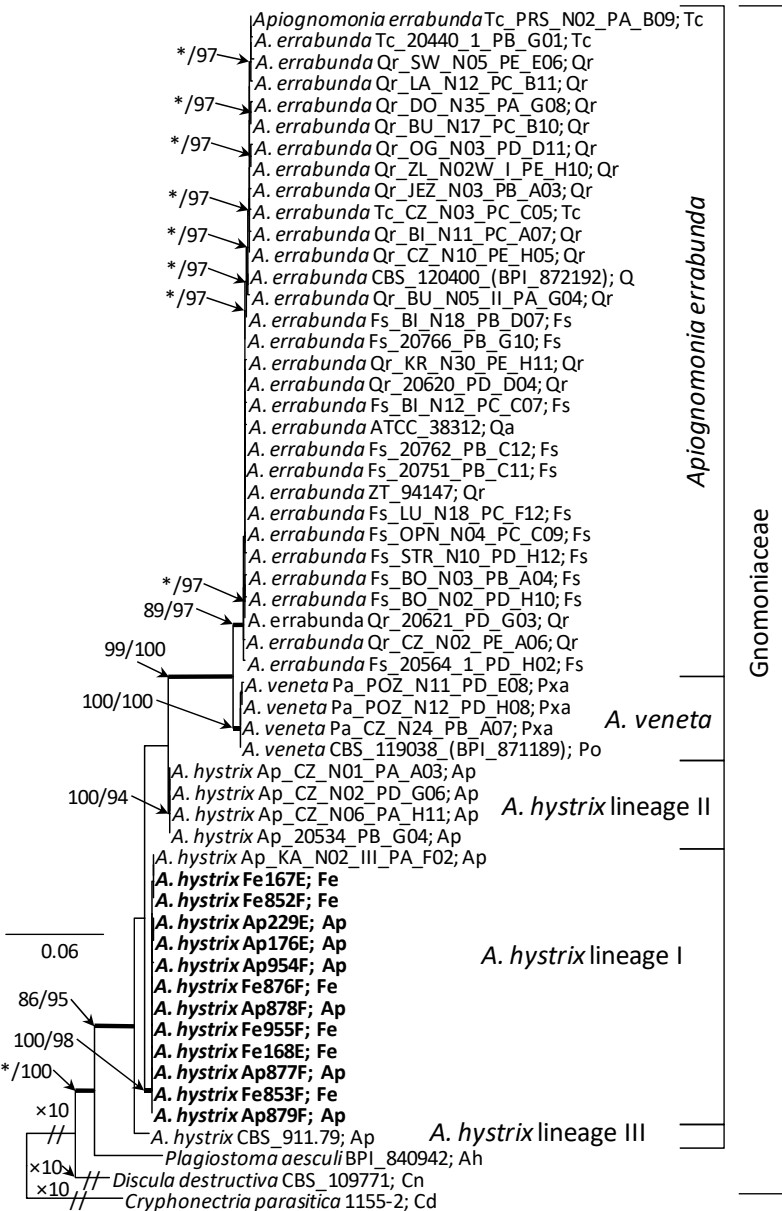

**Figure 6.** Phylogram obtained in maximum likelihood (ML) analysis of the combined ITS-ACT-CAL datasets of the *Apiognomonia* spp. and selected species from Gnomoniaceae. Sequences obtained in this study are marked in bold type. The bootstrap values ≥ 75% for ML and maximum parsimony (MP) analyses are presented at the nodes using ML/MP pattern. Bold branches indicate posterior probability values ≥ 0.95 obtained in Bayesian Inference (BI) analysis. * Bootstrap values < 75%. The tree is drawn to scale (see bar) with branch length measured in the number of substitutions per site. *Cryphonectria parasitica* 1155-2 represents the outgroup. The abbreviations used after the semicolon indicate the host plants for isolates: Ah—*Aesculus hippocastanum*, Ap—*Acer pseudoplatanus*, Cd—*Castanea dentata*, Cn—*Cornus nuttallii*, Fe—*Fraxinus excelsior*, Fs—*Fagus sylvatica*, Po—*Platanus occidentalis*, Pxa—*Platanus × acerifolia*, Q—*Quercus* sp., Qa—*Quercus alba*, Qr—*Quercus robur*, Tc—*Tilia cordata*.

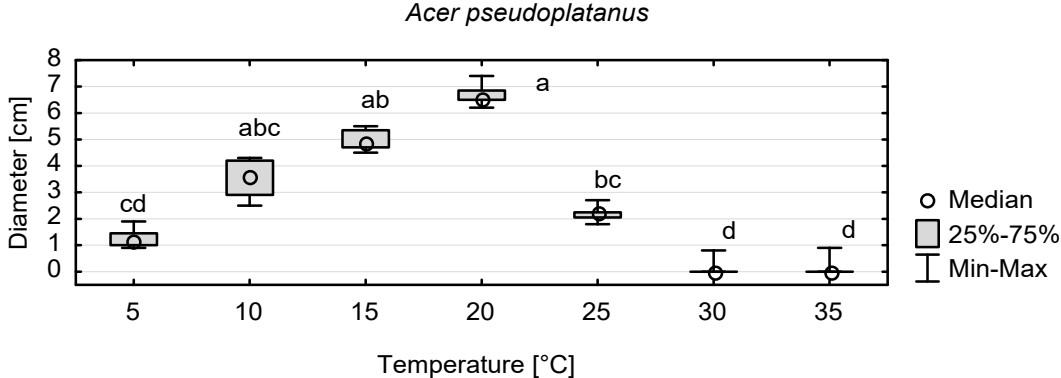

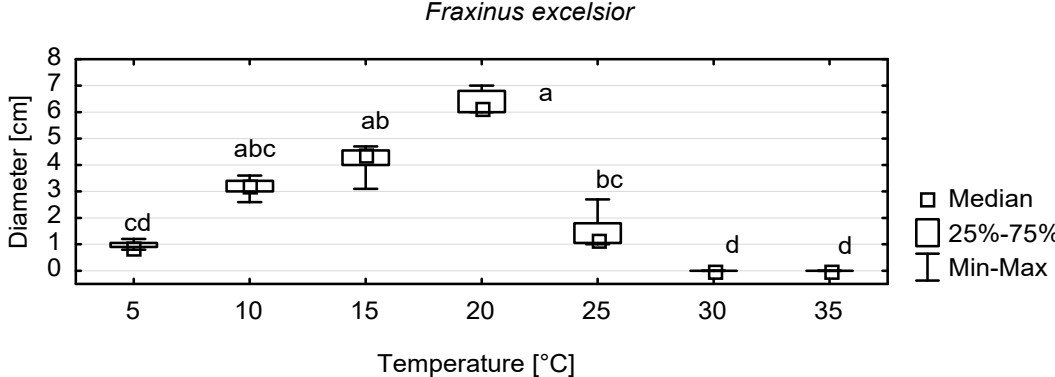

**Figure 7.** Diameters of *Apiognomonia hystrix* colonies isolated from *Acer pseudoplatanus* and *Fraxinus excelsior* after 10 days of growth at various temperatures. Values indicated with different letters have been determined significantly different at $\alpha = 0.05$ in Kruskal–Wallis test.

### 3.4. Pathogenicity of Apiognomonia hystrix

Among 48 *A. pseudoplatanus* petioles inoculated with *A. hystrix*, 41 developed necroses after 8 weeks, 21 necroses caused by isolates originating from sycamore and 20 necroses were caused by isolates from European ash (Table 4). In most of the lesions, necrotic tissues were light beige with grey or purple-grey discolouration at the necrosis margin (Figure 8a–d). Seven of the inoculated petioles were already wilted or had dropped from the tree at the evaluation date. Thirty-seven out of 41 petioles with necroses also developed *A. hystrix* conidiomata, although for 6 petioles, the conidiomata were present only within the inoculation wound. On the dropped leaves, *A. hystrix* conidiomata were generally produced throughout the entire discoloured parts of petiole (Figure 8e). In 5 out of 7 inoculated petioles on which necroses did not develop, the incised fragment turned noticeably lighter in colour and harboured *A. hystrix* conidiomata (Figure 8f). The incised fragments on two remaining necrosis-free petioles were dead and without any conidiomata (Figure 8g).

The experiment involving inoculation of leaf petioles proved that the *A. hystrix* isolates used in the test had the ability to cause necroses, which were statistically longer than the control (Figure 9). The size of necroses on *A. pseudoplatanus* caused by particular isolates varied, but most of these differences were not statistically significant. Two exceptions from this pattern were isolates Fe168E and Ap176E, which had necroses that were statistically significantly larger than those caused by both the isolates Ap954F and Fe876F and the isolate Fe876F (Table 4).

**Table 4.** Number of *Acer pseudoplatanus* (Ap) and *Fraxinus excelsior* (Fe) petioles with necrotic lesions and the lesion size, eight weeks post inoculation with *Apiognomonia hystrix*.

| Isolate | Inoculated Host | Inoculated Petioles (n) | Occurrence of Longitudinal Necrosis (n) | Mean Length of Necrosis (mm) | Median (mm) [1] | Standard Deviation | Standard Error of Mean |
|---|---|---|---|---|---|---|---|
| | | | Isolation source—*Acer pseudoplatanus* | | | | |
| Ap176E | Ap | 6 | 6 | 57.5 | 56.5 [ab] | 12.5 | 5.1 |
| | Fe | 6 | 0 | 0.0 | 0 | 0.0 | 0.0 |
| Ap229E | Ap | 6 | 5 | 37.8 | 35.5 [abc] | 24.0 | 9.8 |
| | Fe | 6 | 0 | 0.0 | 0 | 0.0 | 0.0 |
| Ap877F | Ap | 6 | 5 | 23.0 | 24.5 [abc] | 13.2 | 5.4 |
| | Fe | 6 | 0 | 0.0 | 0 | 0.0 | 0.0 |
| Ap954F | Ap | 6 | 5 | 22.3 | 17.5 [bc] | 21.6 | 8.8 |
| | Fe | 6 | 0 | 0.0 | 0 | 0.0 | 0.0 |
| | | | Isolation source—*Fraxinus excelsior* | | | | |
| Fe168E | Ap | 6 | 6 | 67.2 | 64 [a] | 11.0 | 4.5 |
| | Fe | 6 | 0 | 0.0 | 0 | 0.0 | 0.0 |
| Fe852F | Ap | 6 | 4 | 26.7 | 28.5 [abc] | 23.7 | 9.7 |
| | Fe | 6 | 0 | 0.0 | 0 | 0.0 | 0.0 |
| Fe876F | Ap | 6 | 4 | 14.5 | 17.5 [c] | 12.0 | 4.9 |
| | Fe | 6 | 0 | 0.0 | 0 | 0.0 | 0.0 |
| Fe955F | Ap | 6 | 6 | 44.8 | 37 [abc] | 21.9 | 8.9 |
| | Fe | 6 | 0 | 0.0 | 0 | 0.0 | 0.0 |
| Control | Ap | 6 | 0 | 0.0 | 0 | 0.0 | 0.0 |
| | Fe | 6 | 0 | 0.0 | 0 | 0.0 | 0.0 |

[1] Values indicated with different letters have been determined significantly different at $\alpha$ = 0.05 in Kruskal–Wallis test. Only *Acer pseudoplatanus* calculations have been presented.

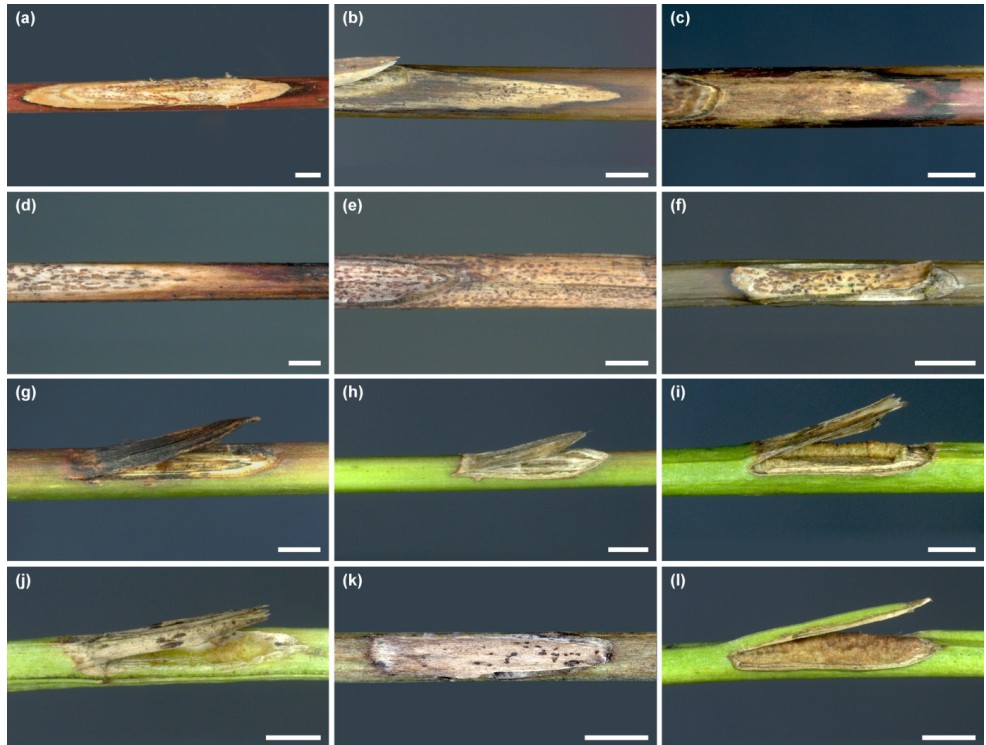

**Figure 8.** Occurrence of necrotic lesions on petioles of *Acer pseudoplatanus* and *Fraxinus excelsior*, eight weeks after wound inoculation with *Apiognomonia hystrix*. *A. pseudoplatanus* (**a–h**): extensive necrosis with light discoloration of necrotic tissues and with conidiomata of *A. hystrix* (**a–d**), petiole of fallen leaf with *A. hystrix* conidiomata covering the entire necrosis surface (**e**), inoculation wound covered with conidiomata, no tissue necrosis within the surrounding area (**f**), inoculation wound without developed conidiomata, no tissue necrosis within the surrounding area (**g**), control petiole without necrotic lesion (**h**); *F. excelsior* (**i–l**): partially closed wound with no discoloration of the surrounding tissue (**i**), wound without callus formation and no necrosis of the surrounding tissue, mature conidiomata of *A. hystrix* on the inoculation wound (**j,k**), control petiole with fully closed wound (**l**); scale bars: (**a–l**) = 2 mm.

None of the 48 inoculated *F. excelsior* petioles developed necrotic lesions (Table 4). On six of the petioles, the inoculation wounds healed completely, and an additional 27 of the petioles had partially healed the inoculation wounds (Figure 8i); no healing of the inoculation wounds was observed in 15 petioles (Figure 8j). *Apiognomonia hystrix* conidiomata, within the incised petiole section, were observed on 8 *F. excelsior* petioles at the evaluation date, resulting in light-beige discolouration of the dead tissues (Figure 8j,k).

According to the Mann–Whitney U test, the host origin of the isolates did not differentiate them with regard to the ability to cause necroses on petioles of the studied tree species (Figure 10). All the isolates used in the inoculations caused statistically significantly larger necroses on *A. pseudoplatanus* petioles compared to those on *F. excelsior* petioles (Figure 11).

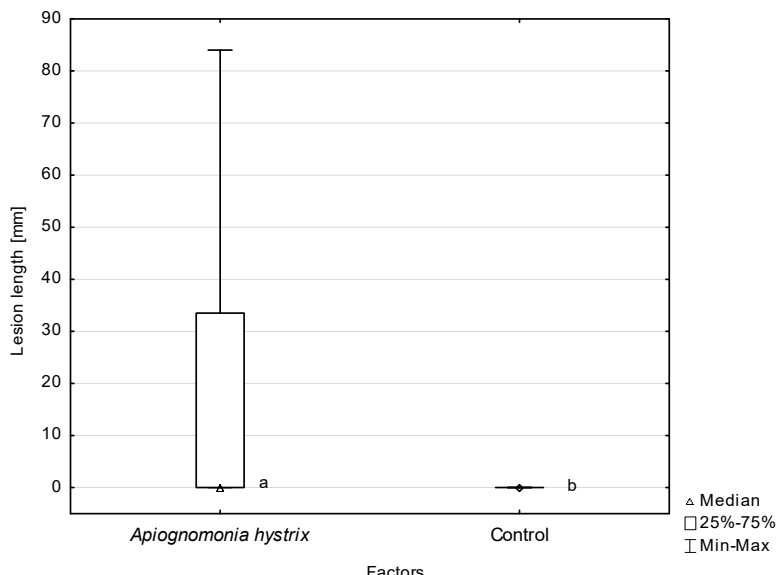

**Figure 9.** Lengths of necrotic lesions on petioles inoculated with *Apiognomonia hystrix* and in the control, eight weeks post inoculation. Values indicated with different letters have been determined significantly different at α = 0.05 in Mann–Whitney U test.

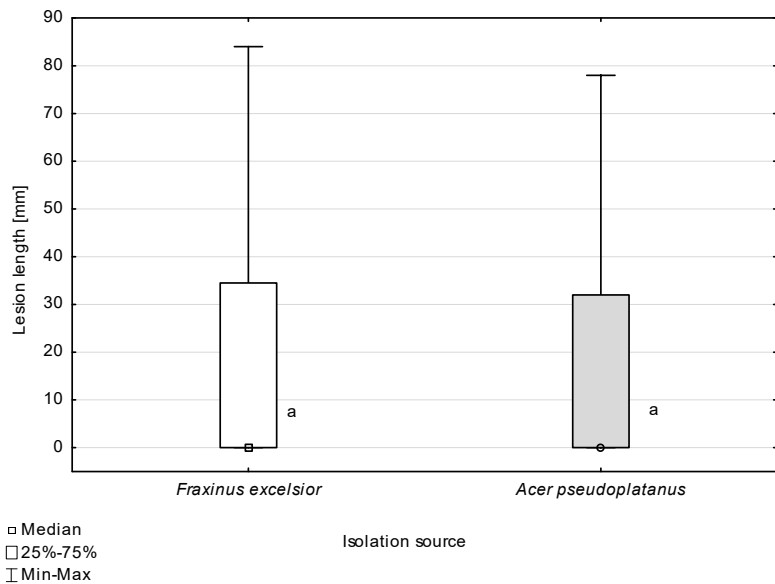

**Figure 10.** Lengths of necrotic lesions on petioles inoculated with *Apiognomonia hystrix* in regard to isolates' host origin, eight weeks post inoculation. Values indicated with the same letters have been determined significantly not different at α = 0.05 in Mann–Whitney U test.

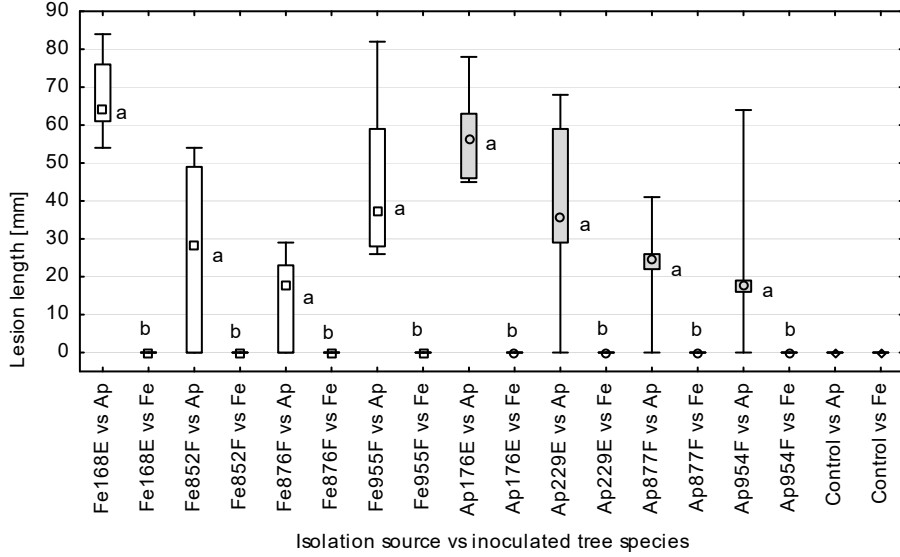

**Figure 11.** Lengths of necrotic lesions on *Fraxinus excelsior* (Fe) and *Acer pseudoplatanus* (Ap) petioles inoculated with *Apiognomonia hystrix* isolates originating from *Fraxinus excelsior* and *Acer pseudoplatanus*, eight weeks post inoculation. Values within groups (created by combining the data for the same isolates) marked with different letters indicate statistically significant differences between host species according to the Mann–Whitney U test with significance threshold $\alpha = 0.05$.

### 3.5. Controls and Re-Isolations

None of the control petioles developed necroses (Figure 8h,l). The control inoculation wounds on 3 and 4 petioles of *A. pseudoplatanus* and *F. excelsior*, respectively, healed completely by the evaluation date; for the rest of the control wounds, the healing was incomplete. No *A. hystrix* cultures could be isolated from the control petioles regardless of the host tree. *Apiognomonia hystrix* cultures were successfully reisolated from 91.7% of the inoculated petioles of *A. pseudoplatanus* and from the inoculation wounds from 18.8% of the inoculated *F. excelsior* petioles. Numerous fungi other than *A. hystrix* were isolated from the control petioles, including species of the genera *Alternaria*, *Colletotrichum*, *Fusarium* and *Nemania*.

## 4. Discussion

### 4.1. Morphological and Phylogenetic Aspects

The *A. hystrix* lifecycle includes the production of conidiomata and perithecia [5,13,17]. Both of these fructification types were detected in this study on *A. pseudoplatanus* and *F. excelsior*. *Apiognomonia hystrix* produces 2-celled conidia, which immensely helps to distinguish it from other species, such as *D. umbrinella* (teleomorph *A. errabunda*) or *D. campestris*, of which the conidia are 1-celled [17,22,67,68]. A teleomorph of another *Apiognomonia* species occurs on dead *Acer* spp. petioles, i.e., *A. acerina*. This species is, however, unique in having ascospores wider than 3.5 µm, which makes it easy to distinguish from *A. hystrix* [8,69]. According to Wulf [5], the culture-grown conidia of *A. hystrix* do not differ from those produced in conidiomata on natural substrate, reaching a size of 9–15 × 2–3.5 µm. Our observations indicate, however, that the diversity of conidial size in vitro is greater than that of conidia produced in situ on dead leaf tissues. Similar differences were indicated by Sieber et al. [70]. On the other hand, the ability of *A. hystrix* to produce microconidia has been reported only infrequently [13]. In this study, we observed microconidia production on both *A. pseudoplatanus* leaves and in cultures, for both maple- and ash-originating isolates. Monod [13] describes microconidia production in some cul-

tures acquired from leaves but not in cultures originating from maple shoots. Our results indicate that the occurrence of microconidia depends strongly on the culture age, as these structures are increasingly more frequent in older cultures. Bacilliform or club-shaped microconidia of *A. hystrix* differed significantly from the microconidia shape reported for *A. veneta*, which are subglobose with diameters of less than 3 μm [19].

The studied *A. hystrix* isolates showed significant variation in structure and colour, even when cultured under the same temperature and light conditions. In some cultures, the occurrence of brown or black patches merging with each other was observed. The same culture morphology was reported by Monod [13], who showed that some sections of *A. hystrix* colonies on MEA medium could become melanized. Similarly, Sieber et al. [70] indicated the occurrence of black patches in cultures of *A. hystrix* and significant morphological differences between isolates originating from Canada and Switzerland. Our results and the results obtained by Wulf [5] agree on the optimal-growth temperature of *A. hystrix* mycelium that has been estimated at 20 °C. Both studies indicate that the growth of *A. hystrix* colonies at 25 °C is very limited and practically stops at higher temperatures. This suggests that the expected climate change may render the growth conditions for the species less favourable in the near future [71,72].

Results of phylogenetic analyses indicate that *A. hystrix* isolates are genetically variable, sufficiently so to distinguish three lineages. All the isolates generated in this study belonged to linage I together with isolates acquired by Sogonov et al. [8]. Linage II grouped *A. hystrix* isolates originating from *A. pseudoplatanus* were acquired during the analyses of Boroń et al. [21]. Whereas lineage III comprised sequences for isolates CBS 911.79; CBS 910.79, both acquired by Monod [13], and the isolate CBS 109759 generated by Green and Castlebury [73]. These results suggest that *A. hystrix* is not a monophyletic species. A similar pattern of sequence diversity was revealed for *Apiognomonia* isolates originating from *A. pseudoplatanus*, *Fagus sylvatica* and *Tilia cordata* in the analyses of Boroń et al. [21], which involved the same set of barcode regions as adopted in our study. Based on their three-gene phylogeny, the authors suggest that such a diversity pattern may indicate a substantial level of inter-host group recombination and/or occurrence of interspecies hybrids between *A. hystrix* and *A. errabunda*.

*4.2. Apiognomonia hystrix: Host Plant Spectrum and Disease Symptoms*

Host specialization may be qualitative, characterized by the complete inability of a pathogen species to infect many hosts, or quantitative, where pathogens have a lower performance on certain hosts [74]. This is a very important distinction for *A. hystrix*, as the species occurs on various hardwoods, but only some of these trees are infected with detrimental effects on their health. In many other species, the fungus occurs only occasionally, without any significant impact [2–6,8,9,13,15,27,75]. In this study, we demonstrated that the occurrence of *A. hystrix* is high in necroses surrounding *Da. vitrina* galls, which supports results obtained by other authors [5,6]. Kowalski [27] showed that *A. hystrix* frequently occurs in necroses associated with *Dr. glutinosa* galls, and this result was confirmed in the present study. Both gall midges occur throughout most of the distribution range of their host, *A. pseudoplatanus*, and are serious pests on maple that are capable of causing outbreaks [46,47,76]. Certain species of endophytic fungi that occur in the leaves but are dormant can be stimulated to pathogenic development by the activity of gall insects [77,78]. Such a phenomenon may be observed in other plant species, including beech leaves where developing *Mikiola fagi* or *Hartigiola annulipes* galls are frequently accompanied by *Apiognomonia errabunda* or oak leaves with *Neuroterus numismalis* galls where a related species, *A. quercina*, occurs [6,77].

Thus, based on our results, it may be hypothesized that the occurrence of galls caused by *Da. vitrina* and *Dr. glutinosa* midges may increase the prevalence of *A. hystrix*. Damage to maple leaves resulting from midge activity can occur as soon as July and facilitates the production of *A. hystrix* conidiomata. Thus, germinable *A. hystrix* conidia are produced

in very high numbers with the potential to spread the species within the same growing season either on leaves and on twigs of *A. pseudoplatanus* as well as on other plant species.

Our results proved that the activity of *A. hystrix* on *A. pseudoplatanus* leaves is not restricted to the part of the leaf blade previously affected by gall midges. This is based on the observation of large necroses, significantly exceeding the size of galls [46,47,76], as well as the occurrence of anthracnose-like vein-associated necroses. This latter type of necrosis did not differ from the symptoms produced by *A. veneta*, *A. errabunda* or *A. erythrostoma* on leaves of *Platanus*, *Fagus* or *Prunus*, respectively [20,31,34,79]. Our results show that *A. hystrix* may be considered an opportunistic pathogen that takes advantage of damaged plant tissues. They are also largely consistent with some previous authors' suggestions that *A. hystrix* may be a causal agent of round or irregularly shaped leaf spots on sycamore leaves [13,34,35].

*Fraxinus excelsior* is a known host for *A. hystrix*, as the fungus has been previously isolated from its shoots and living symptomless leaves [37–39], as well as from its dead overwintered petioles [8]. However, all these reports describe only sporadic detection of *A. hystrix*. Our results show that *F. excelsior* in Poland is infected by *A. hystrix* as well, and similarly to other countries, its frequency on this host is very limited. *A. hystrix* host-fungus relationships with *F. excelsior* were similar to those observed in *A. pseudoplatanus*. The fungus was detected as an endophyte in ash leaves and shoots, as a saprotroph on dead petioles and within necrotic tissues surrounding *Da. fraxinea* galls, which are inconspicuous in the form of circular blisters with irregular rounded openings [47].

It is evident, based on the results of mycological investigations involving *A. hystrix* occurrence performed to date, that its occurrence on hosts other than *Acer*, e.g., *Populus tremula*, *Carpinus betulus*, *Fagus sylvatica*, *Ulmus scabra*, *Prunus domestica*, *P. padus* or *Sorbus aria*, is not much higher than the occurrence of *A. hystrix* on *F. excelsior* [13].

*4.3. Pathogenicity of Apiognomonia hystrix*

Many endophytic fungi may become pathogenic to host plants under stress conditions, resulting in disease symptoms [80–82]. This makes investigations of *F. excelsior* and *A. pseudoplatanus* endophytes relevant, as both tree species have recently experienced adverse growth conditions. *Fraxinus excelsior* has been attacked throughout most of its natural range by *Hymenoscyphus fraxineus* for over two decades, making it vulnerable to other pathogens [49]. *Acer pseudoplatanus*, on the other hand, suffers from adverse abiotic stresses due to changing climatic patterns and is increasingly attacked by opportunistic pathogens and wound fungi [25,26], as well as by alien pathogens introduced, either recently or in previous decades, to Europe, such as *Cryptostroma corticale* or *Eutypella parasitica* [83–85].

In this study, we demonstrated a significant level of pathogenicity of *A. hystrix* towards *A. pseudoplatanus*. In total, *A. hystrix* isolates induced tissue necroses on 85.4% of inoculated petioles, with the average necrosis length for particular isolates tested ranging from 14.5 to 67.2 mm after eight weeks. For comparison, the average length of necroses caused in a similar setting by highly pathogenic *H. fraxineus* on *F. excelsior* ranged from 17 to 166 mm [65]. Notably, the average length of necroses caused by *A. hystrix* isolates originating from *A. pseudoplatanus* did not differ statistically from the length of necroses caused by *F. excelsior*-originating isolates, indicating no differences in strain virulence related to their host origin. The opposite behaviour was observed for *Discula umbrinella* cultured from leaves of *Quercus alba* and *Q. rubra*. Seedlings of both oak species inoculated with these isolates developed milder disease symptoms if the isolate originated from *Q. rubra* [86]. This situation may be due to the occurrence of host selective groups within *D. umbrinella*, evidenced by studies of host-related pectic enzyme patterns [87]. In addition, appressorium or halo formation has been observed only on the host surface and never on other substrates, suggesting the host-specific induction of penetrating structures [68].

None of the inoculated *A. pseudoplatanus* petioles developed necroses reaching the petiole base by the evaluation date, nor did the *A. hystrix* mycelium reach the petiole base. This includes even the prematurely dropped leaves. Conversely, *A. veneta* mycelium

on inoculated *Platanus acerifolia* leaves was able to overcome the leaf-shoot barrier and grow into twigs, which resulted in the development of cankers [88]. This is the exact way in which *H. fraxineus* infects *F. excelsior* shoots and twigs, causing the serious disease of European ash known as ash dieback [89]. Our pathogenicity test involved *A. hystrix* inoculation in artificially wounded leaf petioles, whereas the tests described by Milne and Hudson [90] included *A. veneta* inoculations within *P. acerifolia* leaf blades and veins, either artificially wounded or intact. As a result, the leaf blight symptoms on *P. acerifolia* developed only where leaf veins had been wounded. The pathogen appeared unable to establish a successful infection via an intact leaf surface or wound in the lamina [90]. Experiments performed by other authors also indicated that wounding seems to be a prerequisite for infection of plane trees by *A. veneta*. These results seem to be a good comparison for the results obtained in our experiments, especially since *A. veneta* infections may even reach epidemic levels [8,33,79]. Therefore, these data indicate the great importance of factors causing damage of plant tissue for the infection rate of *Apiognomonia* species. These are, among others, gall causing midges.

All studied isolates had the potential to cause necroses of sycamore tissues. However, five of the isolates colonized only in the direct proximity of the inoculation wounds (either on one or on both inoculated petioles), and no colonization of adjacent tissues or subsequent necrosis development was observed in these cases. The situation may be due to differences in host response between particular saplings used in the pathogenicity test [91]. Another reason may involve the position of particular leaves within the crown, especially in regard with its insolation, which determines the leaf's construction cost (g glucose g$^{-1}$) and its chemical composition and morphology [92,93]. According to the study of Unterseher et al. [23] on the infection patterns of fungal endophytes, *A. hystrix* occurred almost exclusively in the understorey, contrary to some other endophytes, whose occurrence within the fully insulated areas of the crown was more frequent. Furthermore, the impact of the buffering capacity of sycamore tissues on the colonization success of *A. hystrix* cannot be ruled out, as according to Zimmermann [24], *A. hystrix* causes strong acidification of the medium in vitro. Therefore, the host plant, by preventing *A. hystrix* from adjusting its tissue pH to the level required by the fungus, may limit the colonization extent [94]. Such a phenomenon has been previously described for pathogens of agricultural crops [95]. Limited colonization observed on some petioles, confined only to direct proximity of inoculation wounds, may be an equivalent of the situation observed on leaves, where *A. hystrix* causes necroses in the direct proximity of galls without widespread colonization of the surrounding leaf blade.

Based on the results obtained so far, it can be concluded that sycamore leaves are more prone to develop visible symptoms of *A. hystrix* infection than the tissues of its trunks. Gregory [25] was able to isolate *A. hystrix* cultures from areas of bark necroses within trunks of *A. pseudoplatanus*, which developed after a particularly dry summer. However, the author remarked that *A. hystrix* is able to cause only limited necrotic extensions of physical bark injuries on healthy sycamore trees and should be treated as an opportunistic pathogen. A certain level of *A. hystrix* pathogenic potential towards *A. macrophyllum* is indicated in the results of Sieber et al. [4]. The authors observed an inhibiting effect of the fungus on *A. macrophyllum* callus in dual cultures in vitro, ultimately resulting in overgrowth and death of the callus.

Based on the results obtained in this study, it may be concluded that *A. hystrix* does not show pathogenic potential towards *F. excelsior*. The same results show, however, that the fungus is able to colonize physically damaged ash tissues, together with production of mature conidiomata. The results of our pathogenicity test support the in vivo observation in this regard. They explain the lack of anthracnose-like symptoms caused by *A. hystrix* infections on leaves of *F. excelsior*, similar to those observed on sycamore. A factor limiting fungal infections of ash species native to Europe may be their ability to produce some metabolites possessing antifungal properties [96]. Moreover, the leaves of *F. excelsior* proved to have higher concentrations of total phenols and flavonoids than their bark tissues [97].

Many studies have demonstrated that phenolics are good indicators of pathogen resistance in trees, although some fungi may be able to utilize carbon-based phenolics as a source of energy [91].

Based on the data acquired to date, it is impossible to state definitively that *A. hystrix* does not cause disease symptoms in European ash, while the detrimental effect of the fungus on sycamore can be aptly summarized using the classification proposed by Burdon [98]. Between three classes of pathogens determined based on their qualitative negative effect on hosts, *A. hystrix* can be included among debilitating pathogens, which cause variable reductions in plant growth through discrete lesions that individually have small effects on plant fitness [98].

## 5. Conclusions

Our results confirmed that *Apiognomonia hystrix* on living leaves of *Acer pseudoplatanus* frequently co-occurs with gall causing midges *Dasineura vitrina* and *Drisina glutinosa*. They also show that the fungus affects the size of gall associated necroses; the *A. hystrix* colonization resulted in statistically significantly larger necrotic lesions. *Apiognomonia hystrix* conidiomata on sycamore leaves occurred frequently within vein-associated necroses (anthracnose). Whereas *A. hystrix* perithecia were frequently observed on previous year sycamore petioles. Our results also confirmed *F. excelsior* as a host for *A. hystrix*. The fungus occurred on both *F. excelsior* leaflets in association with *Dasineura fraxinea* galls and, infrequently, on overwintered petioles. An important observation pertains to variability of *A. hystrix* colonies in terms of their morphology and tendency to produce conidiomata, containing both macro- and microconidia. Phylogenetic analysis suggests that *A. hystrix* as a taxon is not monophyletic, but all the isolates generated in this study, originating from *A. pseudoplatanus* and *F. excelsior*, were grouped in a single clade (linage I). Pathogenicity tests in situ involving artificial inoculations of petioles showed a significant level of *A. hystrix* pathogenicity toward *A. pseudoplatanus* and its lack of pathogenicity toward *F. excelsior*. In general, our results indicate the potential of *A. hystrix* as a damaging agent, able to manifest itself as a result of the weakened health condition of trees due to other factors.

**Supplementary Materials:** The following supporting information can be downloaded at: https://www.mdpi.com/article/10.3390/f13010035/s1, Figure S1. Location of forest sites, in which living symptomatic leaves of *Fraxinus excelsior* (square) and *Acer pseudoplatanus* (triangle) have been collected for the analysis of *Apiognomonia hystrix* occurrence, collection years 2012–2017. 1—Gryfice, 2—Krzeszowice, 3—Ojców, 4—Miechów, 5—Kraków, 6—Nowy Targ, 7—Stary Sącz. Figure S2. Locations of sampling sites of overwintered leaf petioles of *Acer pseudoplatanus* (triangle) and *Fraxinus excelsior* (square). Full-coloured markers indicate the occurrence, and outline markers indicate the lack of occurrence of *A. hystrix* ascomata. Localities: 1—Suwałki-Stara Hańcza, 2—Suwałki-Szeszupka, 3—Mikołajki, 4—Miłomłyn, 5—Trzęsacz, 6—Kowary, 7—Oława-Jelcz, 8—Puławy, 9—Jędrzejów, 10—Świerklaniec, 11—Prudnik, 12—Rybnik, 13—Krzeszowice-Dubie, 14—Ojców, 15—Domiarki, 16—Andrychów-Brody, 17—Myślenice, 18—Gorce, 19—Stary Sącz-Przysietnica, 20—Krynica-Kopciowa, 21—Dynów-Dąbrówka, 22—Rozpucie, 23—Jabłonki.

**Author Contributions:** Conceptualization, T.K. and P.B.; methodology, T.K. and B.G. (mycological aspects and pathogenicity test) and P.B. (molecular and statistical aspects); investigation, T.K., P.B. and B.G.; formal analysis, T.K. and P.B.; data curation, T.K. and P.B.; writing—original draft preparation, T.K. and P.B.; writing—review and editing, T.K., P.B. and B.G.; software, P.B.; supervision, T.K. and P.B.; visualization, P.B.; project administration and funding acquisition, T.K. All authors have read and agreed to the published version of the manuscript.

**Funding:** This work was supported by project No. 2016/21/B/NZ9/01226, financed by the National Science Centre, Poland. The article processing charge was financed by a subvention from the Polish Ministry of Science and Higher Education for the University of Agriculture in Krakow for 2021.

**Institutional Review Board Statement:** Not applicable.

**Informed Consent Statement:** Not applicable.

**Data Availability Statement:** The data presented in this study are available in supplementary material.

**Acknowledgments:** The authors express their thanks to M. Skrzypczyńska for her help in species identification of gall midges and to Paweł Szczygieł, the Head of the Stary Sącz Forest District for his help in conducting the field experiments. The authors also thank the anonymous reviewers for their valuable comments.

**Conflicts of Interest:** The authors declare no conflict of interest.

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
