# Peer review of "The Occurrence of Apiognomonia hystrix and Its Pathogenicity towards Acer pseudoplatanus and Fraxinus excelsior under Field Conditions"

_forests, doi:10.3390/f13010035_

Round 1
Reviewer 1 Report
The manuscript studies the potential pathogenicity, associated symptoms, and morphological characteristics of A. hystrix on A. pseudoplatanus and F. excelsior by inoculation in the field. It is well written, easy to read and has a logical flow. The Morphological and Phylogenetic Analyses, symptoms evaluations and control in Methods are standard, and results are presented properly. I enjoyed reading it and specially enjoyed the discussion part where the authors have critically analysed the finding.
In the Discussion part, the study findings are properly compared with available studies such as Seiber et al. and Monod. I found some statement interesting that may need further studies such as “it may be hypothesized that the occurrence of galls caused by Da. vitrina and Dr. glutinosa midges may increase the prevalence of A. hystrix.” However, there are statement that I am not finding them useful or adding any value to the text rather they add to the confusion. Such as” The studied A. hystrix isolates showed significant variation in structure and color, even when cultured under the same temperature and light conditions, which may be at least partially due to their different origins regarding location, host species and plant organ”. Is this really a good justification? If not remove. I have highlighted some text in the pdf file but no specific comments.

Author Response
We would like to thank for the time and effort given to review our manuscript and for comprehensive comments and suggestions. We believe that the manuscript has been significantly improved, thanks to Your valuable contribution. We express our gratitude to both Reviewers in the manuscript in Acknowledgements. We took into account Your suggestions and answer all your questions.
Reviewer 2 Report
This is a very interesting paper and I have done only a few comments. There are no problems with language or terminology. The results were presented very well, and I have only a few minor comments to make a little bit improved. I am sure that all points mentioned below will be able to make better easily.
Introduction:
Basically, I`d suggest adding the very short description of Dasineura fraxinea and Drisina glutinosa
Line 99 – add the reference for your studies on F. excelsior endophytes and mycobiota associated with ash leaf petioles in the litter.
M&M
Line 152 – add temperature and light/darkness for culturing
Line 231- Please explain more detail about similarity, e.g. similarity to taxon level 98–100 %, similarity to genus level 94–97 %
Discussion:
Line s 679-681 - This is a very interesting piece of information. This possibility should be discussed more in-depth. What was the fate of the inoculation tests after the end of the experiment? Did the disease show up later after 8 weeks in 48 inoculated F. excelsior? Please add if you may provide such information
Author Response
We would like to thank for the time and effort given to review our manuscript and for comprehensive comments and suggestions. We believe that thanks to Your valuable contribution the manuscript has been significantly improved. We express our gratitude to both Reviewers in the manuscript in Acknowledgements. We took into account Your suggestions.
Remark 1
Introduction: Basically, I`d suggest adding the very short description of Dasineura fraxinea and Drisina glutinosa
Answer 1
we have added the text: line 104-113.
Remark 2
Line 99 – add the reference for your studies on F. excelsior endophytes and mycobiota associated with ash leaf petioles in the litter.
Answer 2
We have added two references: line 99-100
Kowalski, T.; BilaÅ„ski, P. Fungi detected in the previous year’s leaf petioles of Fraxinus excelsior and their antagonistic potential against Hymenoscyphus fraxineus. Forests 2021, 12, 1412. https://doi.org/10.3390/ f12101412
Bilański, P.; Kowalski, T. Fungal endophytes in Fraxinus excelsior petioles and their in vitro antagonistic potential against the ash dieback pathogen Hymenoscyphus fraxineus. Microbiol. Res. 2022. accepted.
Remark 3
Line 152 – add temperature and light/darkness for culturing
Answer 3
Line 179 We have added this data:
All isolations and subsequent in vitro culturing steps were conducted on 2% malt extract agar medium (MEA: 20 g l-1 malt extract, Difco, 15 g l-1 agar; Difco, Sparks, MD, USA, supplemented with 200 mg l-1 tetracycline; Tetracyclinum TZF Polfa, Poland) in Petri dishes, at 20°C in the dark.
Remark 4
Line 231- Please explain more detail about similarity, e.g. similarity to taxon level 98–100 %, similarity to genus level 94–97 %
Answer 4
Line 243-244 We have added the following text: For species and genus level, the accepted similarity sequences threshold was 99–100% and 91–98%, respective.
Remark 5
Line 679-681 - This is a very interesting piece of information. This possibility should be discussed more in-depth. What was the fate of the inoculation tests after the end of the experiment? Did the disease show up later after 8 weeks in 48 inoculated F. excelsior? Please add if you may provide such information
Answer 5
we added the text line 708-710: Therefore these data indicate the great importance of factors causing damage of plant tissue for the infection rate of Apiognomonia species. These are, among others, gall causing midges.
Unfortunately after 8 weeks the inoculated petioles were harvested and used for reisolation. So in order to gain further information the experiment would need to be repeated with the use of other methodology.